# Effects of Exercise Intervention on Students’ Test Anxiety: A Systematic Review with a Meta-Analysis

**DOI:** 10.3390/ijerph19116709

**Published:** 2022-05-31

**Authors:** Xueyan Zhang, Wenhao Li, Jinghao Wang

**Affiliations:** School of Physical Education, South China Normal University, Guangzhou 510006, China; 2020021094@m.scnu.edu.cn (X.Z.); 2021021119@m.scnu.edu.cn (W.L.)

**Keywords:** exercise, students, test anxiety, meta-analysis, physical activity

## Abstract

This paper aims to evaluate the effects of exercise intervention on students’ test anxiety with a meta-analysis to serve as a reference to further relevant research. CNKI, Wanfang Data, VIP, The Cochrane Library, PubMed, Web of Science, Scopus, Embase, and EBSCOhost databases were electronically searched from inception to August 2021 in order to identify randomized and non-randomized controlled trials on exercise for treating students’ test anxiety. Two researchers independently screened the study, extracted data, and evaluated the quality of the studies. Meta-analysis was carried out using Review Manager 5.4 software. Among the 4115 articles retrieved, 16 were qualitative analyses, 15 of which were included in the meta-analysis. The meta-analysis showed that exercise intervention effectively alleviated students’ test anxiety when compared with the blank control group. Subgroup analysis showed that a single exercise intervention lasting 10~15 min did not relieve students’ test anxiety. The different scales used in the study are the source of inter-study heterogeneity. In conclusion, twenty minutes of aerobic exercise at any intensity performed two to three times a week for at least four weeks can reduce test anxiety levels. However, this is the minimum level and students should not be limited to the minimum amount of exercise when reducing test anxiety. In addition, exercise combined with psychotherapy can reduce students’ test anxiety better than exercise intervention alone. Future studies should explore the minimum intervention period and whether the shorter intervention period can quickly and effectively reduce students’ test anxiety.

## 1. Introduction

Test anxiety is a serious problem for many students. Chinese scholars have conducted many studies on test anxiety among Chinese students. The research shows that test anxiety is widespread among students, which increases as they get older [1,2,3]. Reducing the level of test anxiety has always been an important issue in the study of test anxiety. 

Test anxiety is a kind of disordered emotion in the examination situation [4]. Test anxiety is a psychological state in which anxiety is the basic characteristic, defense, or escape is the behavior pattern, and different degrees of emotional reaction are produced under the restriction of individual factors in certain examination situations [5].

Caspersen defined exercise as “A physical activity that is planned, structured, repetitive, and purposive in the sense that improvement or maintenance of one or more components of physical fitness is an objective” [6].

With a large population in China, students need to compete with each other to attain a good university or job. Especially in the National College Entrance Examination, Chinese students face great test anxiety.

According to existing research, exercise can improve the level of overall health and life expectancy, help protect against heart disease and obesity, and it could improve the symptoms of depression [7,8]. Exercise is thought to be a stress-relieving behavior, and the stress-buffering hypothesis postulates that participation in regular physical activity buffers the negative health effects of stress [9]. Most of the interventions for test anxiety are behavioral approaches, cognitive approaches, etc. Previous meta-analysis studies have integrated a variety of intervention methods, such as cognitive therapy, behavioral therapy, cognitive behavioral therapy, and skill focus, but did not involve exercise intervention [10,11,12]. Previous meta-analysis has concluded that the most effective treatment was a combination of skill-focused approaches with behavior or cognitive approaches. Drug therapy may have some side effects [13], and psychotherapy requires too long of a wait time, leading to more severe symptoms. Exercise, an inexpensive and self-controlled form of treatment, has been proven to have many physical and psychological benefits. Therefore, exercise may be a surrogate factor for a reluctance to accept medication or psychotherapy. Controlled research of exercise intervention provides evidence of the effectiveness of test anxiety reduction interventions. The prevalence of controlled studies is hampered by small sample sizes, different research environments and conditions, and conflicting results. A meta-analysis addresses these shortcomings by synthesizing existing studies into one comprehensive study.

The purpose of this study is through the meta-analysis to explore the effect of exercise intervention on students’ test anxiety and provide the basis for the clinical application of exercise intervention in students’ test anxiety.

## 2. Materials and Methods

### 2.1. Search Strategy

In this meta-analysis, Chinese and English studies on exercise intervention on students’ test anxiety were searched via computer. Both Chinese and English studies were retrieved because the number of only English studies alone was too small a sample size to be convincing. Thus, both Chinese electronic databases (CNKI, VIP Database, Wanfang Database) and English electronic databases (Embase, EBSCOhost, PubMed, Web of Science, Cochrane Library, and Scopus) were utilized. These databases are authoritative and contain a large number of articles on the topics needed. They are also frequently selected in other articles that have been published. The retrieval date is from the establishment of each database to August 2021. Using the PICOS search tool, the Medical Subject headings (MeSH) contain terms in four areas: population, intervention, outcome, and study design. After several pre-retrievals, it was determined to use the “P+I+O+S” retrieval formula and make appropriate adjustments according to the specific database. Studies were found by searching for “subject words + free words”, using truncation characters and wildcards, and logical operators “or, and” to ensure the comprehensiveness and accuracy of the search. The subject words used were “Students”, “Exercise”, and “Test Anxiety”. Other words were free words. In addition, relevant experimental studies that may be included were obtained by browsing the references of the included study. The types of study searched were randomized controlled trials or non-randomized controlled trials. Search terms included the following: students, primary school students, middle school students, high school students, college students, adolescent, exercise, sports, test anxiety, exam anxiety, exam stress, etc. The terms associated with each group were then listed by the Boolean term “OR” and the resulting three groups were combined using the Boolean term “AND.” Taking PubMed retrieval as an example, see Table 1.

### 2.2. Selection and Exclusion Criteria of Study

Studies which included all of these criteria were selected: (1) The research objects are students, including primary school students, middle school students, college students, and graduate students. (2) The physical activities studied can be individual-based or group-based interventions, long-term or short-term interventions, but excluding non-physical activities such as muscle relaxation, meditation, mindfulness, and other interventions. (3) The control group could be no intervention or any non-physical intervention. (4) The research presentation language should be Chinese or English. (5) The experiment should be a randomized controlled experiment or a non-randomized controlled experiment. 

Studies which included any of the following criteria were excluded: (1) Physical activity interventions for the treatment of mental health problems other than test anxiety. (2) The outcome measures were not related to test anxiety. (3) Review or meta-analysis. (4) No control groups.

### 2.3. Data Extraction and Quality Evaluation

#### 2.3.1. Data Extraction and Processing

Two researchers independently screened studies according to the aforementioned inclusion criteria, and any differences were either resolved through discussion or discussion with a third investigator. The searched studies were imported into NoteExpress study manager for screening. First, duplicate documents were removed. Then, each study was screened by reading the titles and abstracts. Finally, studies that could not be screened according to their titles and abstracts were screened after fully reading the article. Excel spreadsheets were used to extract data from the articles. Two researchers extracted the data independently, including the author, publication time, test groups, and control groups (sample sizes, measurement tools for test anxiety, means, and standard deviations before and after intervention), intervention method, intervention intensity, intervention time, and frequency.

#### 2.3.2. Study Quality Evaluation

The methodological quality of the included studies was independently assessed by two reviewers using the Cochrane Collaboration tool for assessing risk of bias [14] and recorded using Review Manager 5.4 Software (Cochrane, London, UK). Randomized controlled trial articles were evaluated in the following seven areas: Random sequence generation, allocation concealment, blinding of participants and personnel, blinding of outcome assessment, incomplete outcome data, selective reporting, and other bias. Studies were assessed as being at either “low”, “unclear”, or “high” risk of bias across each of these domains.

The risk of bias for non-RCTs was assessed with the methodological index for non-randomized studies (MINORS) [15]. The following 12 aspects were included: A clearly stated aim, inclusion of consecutive patients, prospective collection of data, endpoints appropriate to the aim of the study, unbiased assessment of the study endpoint, follow-up period appropriate to the aim of the study, loss to follow up less than 5%, prospective calculation of the study size, an adequate control group, contemporary groups, baseline equivalence of groups, adequate statistical analyses. There were 0–2 points assigned for each item: 0 points means no report; a score of 1 indicates reporting but insufficient information; a score of 2 indicates reporting and providing adequate information, and the total score is 24. Score of 0–8 is classified as low quality, 9–16 as medium quality, and 17–24 as high-quality study.

### 2.4. Statistical Analysis

Review Manager 5.4 software was used for data analysis. The main results were analyzed for the impact of exercise intervention on test anxiety, and funnel plots were used to look for publication bias. Sensitivity analysis was conducted by “deleting one item of study data”. The results were presented in text only when the deletion of an item affected the results of the study. When different scales were used in each experiment, standardized mean, difference, SMD (standardized mean difference), and 95% confidence intervals (CIs) were selected as the combined statistics. P takes 0.05 as the reference level, and the I^2^ value is used to judge the size of heterogeneity. If *p* ≥ 0.05 and I^2^ ≤ 50%, the statistical heterogeneity between studies is low, and a fixed-effect model is selected. If *p* < 0.05 and I^2^ > 50%, select random effects model for meta-analysis.

According to the Cochrane Handbook [16], when analyzing data from studies with two or more intervention groups, excluding one intervention group will reduce the accuracy of results, and double-counting will mistakenly increase the accuracy of results, both of which are inappropriate.

The handbook also explains that two experimental results can be combined into a group according to the formula, and this paper adopts this method to merge the data. 

Specific methods: The sample size of intervention group 1 was *N*1, the mean was *M*1, and the standard deviation was *SD*1. The sample size of intervention group 2 was *N*2, the mean was *M*2, and the standard deviation was *SD*2. The combined group was *N*1 + *N*2, mean was $\frac{N1M1+N2M2}{N1+N2}$, and the standard deviation was $\sqrt{\frac{(N1-1)SD1^{2}+(N2-1)SD2^{2}+\frac{N1N2}{N1+N2}\left(M1^{2}+M2^{2}-2M1M2\right)}{N1+N2-1}}$. If there is a third group, the two groups combine with the third group, and so on.

## 3. Results

### 3.1. Selection Outcomes

A total of 4115 articles were retrieved, of which 648 were duplicates. A total of 3349 studies were excluded following the title and abstract screening, and 102 studies following the full-text screening. Ultimately, 16 studies were included within the narrative synthesis, 15 of which were included in the meta-analysis. The formulae cannot be collected completely in all of the databases as the search formula was not precisely followed. Instead, the search was expanded to only “test anxiety”, which found four studies that could be included. The study selection process and reasons for exclusion are presented in Figure 1.

### 3.2. The Basic Characteristics of Included Studies

Of the 16 included studies, 12 were randomized controlled trials [17,18,19,20,21,22,23,24,25,26,27,28]. Most of the studies were randomized at the individual level, while only one study [21] was randomized at the cluster level and four were non-randomized controlled trials [29,30,31,32]. Apart from one study [31] that included two exercise intervention groups, the other studies included only one exercise intervention group. All the studies used aerobic exercise as the experimental intervention, and one study [31] also used anaerobic exercise as an intervention. Twelve studies [18,19,20,21,22,23,24,25,29,30,31,32] compared an exercise intervention group with a blank control group, and four studies [17,26,27,28] compared an exercise intervention group with a non-exercise intervention group (e.g., reading, writing, etc.).

Only six studies [17,19,21,24,28,32] reported the exercise intensity to the intervention group of participants, and the remaining 10 studies [18,20,22,23,25,26,27,29,30,31] did not report on the exercise intensity to the intervention group of participants (based on the intervention description and mode of intervention, the researchers made inferences about the possible strength of each intervention).

The minimum total duration of intervention was 10 min in all trials and the maximum duration was 20 weeks. Four studies [17,21,27,28] had only one intervention over the course of the experiment. The number of interventions ranged from 1 to 14 times per week, and most of the research intervention times for 2~3 times a week. The duration of each intervention ranged from 10 to 190 min and most of the intervention therapy was from 30 to 60 min. The details of the included studies are listed in Table 2.

#### 3.2.1. Country

Sixteen included studies were conducted in China (*n* = 8) [18,19,20,22,23,24,29,30], the United States (*n* = 5) [21,27,28,31,32], Germany (*n* = 1) [25], Australia (*n* = 1) [17], and India (*n* = 1) [26].

#### 3.2.2. Study Sample

Sample sizes of the groups ranged from 11 to 275, totaling 1757 people. The mean age of the sample ranged from 10.7 to 21.4 years, the studies included all genders, and eight of the studies included college students [19,20,25,26,27,28,31,32]. Six studies were conducted on middle school students [18,22,23,24,29,30] and two studies were conducted on primary school students [17,21].

#### 3.2.3. Measurement of Study Results

In all the studies, test anxiety was determined by self-reported outcome measures. Two studies used TAI (Test Anxiety Inventory) [24,30]. Five studies used TAS (Test Anxiety Scale) [18,19,20,29,31]. One study used the TAQ (Test Anxiety Questionnaire) [32]. The Multiple Affect Adjective Checklist (MAACL) scale was used in two studies [27,28]. One study used the NA (Negative Affect) scale [25]. One study used CTAS (Children’s Test Anxiety Scale) [21]. A 1–9 Likert scale was used in one study [17]. Two studies used the Test Anxiety Inventory (Zheng Richang) [22,23]. One study used STAI (State Trait Anxiety Inventory) [26].

#### 3.2.4. Intervention Characteristics

The intervention and control groups in each study differed in the type, frequency, and duration of exercise interventions.

All studies used aerobic exercise interventions, except one study that included anaerobic exercise [31]. Three studies included sports games [23,24,29]. Two studies included yoga interventions [26,30]. Two studies included tai chi [18,20]. One study included aerobic dancing [32]. Five studies included jogging or cycling interventions [24,25,27,28,30]. Three studies included various ball games [19,30,31]. One study included physical education [21]. Three studies included psychotherapy combined with exercise intervention [19,22,23]. One study involved physical training (e.g., push-ups, bobby jumps, running in place) [17].

Four studies reported moderate intensity of intervention [19,24,28,32]. Two studies reported moderate to high intensity of intervention [17,21].

Among the 12 long-term intervention studies with significant effects, the most used exercise intervention frequency was 2–3 times/week (7, 58.3%), followed by 1 time/week (3, 25%), 5 times/week (1, 8.3%), and 14 times/week (1, 8.3%).

The most used exercise intervention lasted 30 min (6, 50%), followed by 20 min (3, 25%), 50 min (1, 8.3%), 60 min (1, 8.3%), and 70 min (1, 8.3%).

The most used intervention duration was 8 weeks (4, 33.3%), followed by 7 weeks (2, 16.7%), 5 days (1, 8.3%), 2 weeks (1, 8.3%), 4 weeks (1, 8.3%), 12 weeks (1, 8.3%), and 16 weeks (1, 8.3%).

### 3.3. Assessment of the Risks of Study Bias

Figure 2 shows the bias risk of each item in the 11 included RCT studies in percentage form. There was no significant risk of bias in these studies, and the overall risk of bias was considered moderate. In multiple studies, some areas were rated as having an “unclear risk of bias” because of the lack of detail reported. Two of the papers were rated as high risk in the “blinding of participants and personnel” and the “blinding of outcome data”. This is because the two articles belong to a master’s degree thesis [22,23].

Table 3 shows the quality evaluation scores of the four non-RCTs in the MINORS entries. In the MINORS entries, scores of articles can be included in the specific provisions of more than 13, 0–8 are divided into low-quality, and 9–16 into the medium-quality, 17–24 are divided into high-quality documents. Three of the studies [29,31,32] were of high quality and one was of medium quality [30].

### 3.4. Impacts of Exercise Intervention on Students’ Test Anxiety

Figure 3 is a forest plot examining the efficacy of exercise for the treatment of test anxiety. The meta-analysis of 15 studies showed that exercise interventions had a significant effect on the treatment of test anxiety (SMD = −0.75 (95% CI: −1.15, −0.34), Z = 3.58, *p* = 0.0003; I^2^ = 92%). It shows that exercise can effectively relieve students’ exam anxiety. Among them, one study [21] could not be included in the meta-analysis due to the absence of control group data and baseline differences.

The data in Figure 3 show high heterogeneity (I^2^ = 92% > 50%). Therefore, sensitivity analysis is needed to determine the heterogeneity sources of exercise intervention on students’ test anxiety. Study exclusions were performed one at a time to examine the effect of each study on the overall standardized mean difference. The exclusion of any of the studies did not significantly change the combined SMD, and the estimates in each case were within the confidence range of the overall estimates, suggesting that the results of this meta-analysis were stable and reliable. Subgroup analyses were also performed to explore sources of heterogeneity.

### 3.5. Subgroup Analysis

In order to explore the influence of potential moderator variables on the intervention effect, this meta-analysis selected three basic contents of exercise prescription (exercise intensity, exercise frequency, exercise duration) and other factors as moderator variables for subgroup analysis: intervention method, intervention time, intervention period, and measurement scale. Table 4 summarizes the results of subgroup analyses of the effects of exercise interventions on students’ test anxiety.

#### 3.5.1. Intervention Time

The subgroup analysis of intervention time in this study is shown in Figure 4, including three studies with only one-time intervention which showed no statistical difference (SMD = 0.59, 95% CI [−0.35, 1.52], Z =1.23, *p* = 0.22). One-time exercise intervention lasting 10–15 min did not relieve students’ test anxiety. The duration of the long-term intervention ranged from the minimum of 5 days to 20 weeks, and the time of the intervention ranged from 20 min to 190 min, with an average of 54 min per intervention. Long-term intervention was effective in alleviating students’ test anxiety (SMD = −1.05, 95% CI [−1.43, −0.67], Z = 5.39, *p* < 0.00001). Inter-study heterogeneity did not decrease, so intervention time was not the source of heterogeneity.

#### 3.5.2. Measurement Scale

The subgroup analysis of measurement scale in this study is shown in Figure 5. The subgroup analysis showed that TAI (Z = 3.34, chi^2^ = 0, *p* = 0.95, I^2^ = 0%), TAS (Z = 9.64, chi^2^ = 7.53, *p* = 0.11, I^2^ = 47%), Test Anxiety Inventory (Zheng Richang) (Z = 3.56, chi^2^ = 0.48, *p* = 0.49, I^2^ = 0%), MAACL (Z = 1.07, chi^2^ = 11.2, *p* = 0.0008, I^2^ = 91%).

We found that the heterogeneity of three scales, TAI, TAS, and Test Anxiety Inventory (Zheng Richang), was significantly reduced, while the remaining four studies used different scales. Therefore, the difference of outcome measurement scales selected in the study was the reason for the large heterogeneity among the studies.

#### 3.5.3. Exercise Intensity

The subgroup analysis of exercise intensity in this study is shown in Figure 6. Only three of the 15 studies reported heart rate during exercise, and the intensity was divided into low intensity, moderate intensity, and moderate to high intensity. Among them, except for the medium and high intensity, only one study could not be statistically combined, the low intensity (SMD = −2.35, 95% CI [−3.86, −0.84], *p* = 0.002 < 0.05) and the medium intensity (SMD = −0.45, 95% CI) [−0.86, −0.05], *p* = 0.03 < 0.05) There was a significant difference in exercise intervention. Among them, the low-intensity intervention with the largest effect size (−2.35) had the best effect.

#### 3.5.4. Intervention Method

The subgroup analysis of intervention method in this study is shown in Figure 7 and the intervention methods were divided into an exercise intervention group and an exercise combined psychotherapy intervention group. There was a significant difference in the exercise intervention group (SMD = −0.66, 95% CI [−1.14, −0.19], *p* = 0.006 < 0.05). There are three studies on exercise combined with psychotherapy intervention. Li [23] adopts exercise combined with emotion center therapy, Zhai [19] used physical exercise combined with sand table games, and Gao [22] used physical activity combined with psychological counseling (SMD = −1.13, 95% CI [−1.43, −0.83], *p* < 0.00001), which showed a significant difference. The effect size of the exercise combined with psychotherapy intervention group was higher than that of the exercise intervention group, which indicated that the effect of exercise combined with psychotherapy was better than the exercise intervention alone.

#### 3.5.5. Exercise Frequency

The subgroup analysis of intervention frequency in this study is shown in Figure 8, and the intervention frequency is divided into 1 time per week, 2 to 3 times per week, and >3 times per week. Once a week (SMD = −0.25, 95% CI [−1.15, 0.64], *p* = 0.58 > 0.05) indicated no significant difference; 2–3 times a week (SMD = −0.96, 95% CI [−1.50, −0.41], *p* = 0.0006 < 0.05) and >3 times a week (SMD = −1.35, 95% CI [−2.12, −0.57]), *p* = 0.0006 < 0.05) were all significantly different. Among them, >3 times a week, with the largest effect size (−1.35), had the best effect.

#### 3.5.6. Intervention Period

The subgroup analysis of intervention period in this study is shown in Figure 9, and the intervention period was divided into <4 weeks, 4~8 weeks, and >8 weeks. The category of <4 weeks (SMD = −0.20, 95% CI [−1.26, 0.85], *p* = 0.70 > 0.05), indicated no significant difference; 4~8 weeks (SMD = −0.70, 95% CI [−1.08, −0.32], *p* = 0.0003 < 0.05) and >8 weeks (SMD = −1.87, 95% CI [−3.32, −0.42], *p* = 0.01 < 0.05) were all significantly different. Among them, the intervention with the largest effect size (−1.87) for >8 weeks had the best effect.

#### 3.5.7. Exercise Duration

The subgroup analysis of the duration of a single intervention in this study is shown in Figure 10, where the duration of a single intervention was divided into <20 min, 20~30 min, 31~60 min, and >60 min. The category of <20 min (SMD = 0.59, 95% CI [−0.35, 1.52], *p* = 0.22 > 0.05) had no significant difference; 20~30 min (SMD = −0.78, 95% CI [−1.28, −0.28], *p* = 0.002 < 0.05), 31~60 min (SMD = −1.53, 95% CI [−2.46, −0.60], *p* = 0.001 < 0.05), and >60 min (SMD = −1.17, 95% CI [−1.50, −0.85], *p* < 0.00001) had significant differences. Among them, 31~60 min, with the largest effect size (−1.53), had the best effect.

### 3.6. Publication Bias

The publication bias of the included studies is shown in Figure 11. The funnel plot showed that the graph was relatively symmetrical, and Egger’s test was also used, which indicated that there was no obvious publication bias in the included studies (intercept = 2.25 (95% CI: −5.72, 4), *p* = 0.709).

## 4. Discussion

This article is a systematic review of quantitative data analysis about the current research on the effects of exercise intervention on students’ test anxiety. All eligible randomized and non-randomized controlled trials are published articles. This study did not discuss the content of exercise forms of exercise prescriptions because some of the included studies used a variety of exercise methods for mixed intervention. According to the Cochrane Handbook [16], the repeated calculation will erroneously improve accuracy, which is inappropriate, so they were not used.

The subgroup of intervention time was divided into a one-time intervention group (*n* = 3) and a long-term intervention group (*n* = 12). It was found that one intervention did not relieve test anxiety (*p* = 0.22). Although some studies have shown that a short 5 min walk can improve mental state [33], most studies believe that at least 20–30 min of exercise is required to produce psychological benefits [34], and some studies suggest that 40~60 min work better [35]. Therefore, if the duration of the exercise is less than 20 min, the benefits of the exercise have not yet manifested, and the physical activity has stopped at this time, so there is generally no psychological benefit.

In the subgroup of the measurement scale, it was found that the heterogeneity of studies between the same scales was significantly reduced, so it was judged that the scales selected for different outcome measures in the studies were the reasons for the large heterogeneity between studies. Different items measured by different scales have different corresponding scores, resulting in significant differences in the scores of test anxiety outcome indicators. These factors may lead to large heterogeneity among studies.

In the exercise intervention intensity subgroup, the authors estimated the unreported intensity of studies based on the type and duration of exercise. In the low-intensity group, three articles [18,20,26] did not report the heart rate and exercise intensity during exercise. Rather, they adopted the physical and mental exercise intervention methods of tai chi and yoga, which were classified as low intensity. Among the 11 studies in the moderate-intensity group, only three studies [19,24,32] reported heart rate during exercise, which ranged from 60% to 75% of the maximum heart rate. Apart from one study on medium and high intensity [17], which could not be statistically combined, the results showed that low-intensity and moderate-intensity exercise interventions produced significant differences in test anxiety. The reason why moderate- to high-intensity exercise intervention did not affect test anxiety in this article may be due to only one 10 min exercise intervention. Mavilidi also pointed out in the study that anxiety is a more common and chronic disease. Multiple or higher-intensity exercises may be required [17].

In the subgroup of intervention methods, the results showed that both the exercise intervention group and the exercise combined with psychotherapy intervention group had a significant effect on reducing students’ test anxiety, and that the effect of exercise combined with psychotherapy was better than that of the simple exercise intervention group. Other studies have also proved this [36,37]. The effect of exercise combined with psychological counseling in the intervention of college students’ mobile phone addiction is better than that of simple exercise intervention. In schoolwork, the connection between physical education and psychological education can be strengthened, and physical education teachers and psychological teachers are encouraged to jointly carry out activities to reduce students’ test anxiety.

In the subgroup of intervention frequency, the results showed that exercise 2~3 times a week and >3 times a week can reduce test anxiety, while the six studies using the one time a week group included three studies that only performed a one-time exercise intervention. This is the reason for the insignificant difference, and the other three studies have significant effects. Research on exercise frequency is inconsistent. Some studies suggested that exercising once a month can produce psychological effects [38], while others recommended exercising 3~5 times a week [39]. However, most studies believe that 2~3 times a week of exercise is the most appropriate [18,24,25,26,29,30,32,40].

In the intervention period subgroup, the results showed that there was no significant difference in <4 weeks. While there were significant differences between interventions between 4~8 weeks and >8 weeks, the inclusion of three studies with only a one-time exercise intervention in the <4 weeks group may account for the insignificant difference. Except for the three articles that only had one acute intervention, there was no significant difference. The smallest intervention period was 5 days, followed by 2 weeks, which produced significant differences in reducing students’ test anxiety. Zhu’s study [41] conducted moderate exercise intervention for college students 3 times a week for 5 weeks through the “Keep” app, and the results showed that after 5 weeks of intervention, there was a significant difference in relieving college students’ trait anxiety, which was higher than state anxiety. The degree of reduction is slower, and since test anxiety is a trait anxiety, it may take longer interventions to reduce test anxiety levels. Studies have shown that a 4~6-week exercise intervention period has the best effect on relieving the anxiety of college students [42]. In this study, the intervention period >8 weeks had the greatest effect, but only three studies were included. Fewer conclusions may be unreliable, which is one of the limitations of this meta-analysis.

In the subgroup of single intervention duration, the results showed that there was no significant difference in <20 min. The duration of 20~30 min, 31~60 min, and >60 min had significant differences. The American College of Sports Medicine (ACSM) recommends a duration of 20~60 min [43]. A short exercise duration generally does not produce psychological benefits, because physical activity has ceased before the benefits of exercise appear. However, longer exercises are not always better. Excessive long-term exercise at a certain intensity will bring boredom to the athlete, which is not conducive to mental health, and may also cause fatigue and unnecessary damage [44].

In studies of long-term interventions, both high (such as running, cycling, etc.) or low (yoga, tai chi) exercise intensity had a positive effect on test anxiety. The studies in this meta-analysis found some variability in the time, frequency, and duration of exercise interventions. However, 30 min of exercise 2~3 times a week for 7~8 weeks is the most common exercise program. For healthy people, the American College of Sports Medicine (ACSM) recommends at least 30 min of moderate-intensity aerobic exercise five times a week, or at least 20 min of vigorous-intensity aerobic exercise three times a week [43]. Therefore, it is recommended that students perform 20 min of vigorous-intensity aerobic exercise 2~3 times a week, which can achieve the amount of exercise recommended by ACSM and can also reduce students’ test anxiety. However, in order to further improve physical fitness and reduce the risk of chronic disease and disability, the amount of daily exercise should not be limited to the minimum recommended by ACSM [43]. Exercise prescription for the treatment of test anxiety should follow a step-by-step principle, and the amount of exercise should be adjusted according to the individual’s health status and exercise response.

This meta-analysis shows that exercise intervention have a significant effect on students’ test anxiety and concludes that the lowest exercise intervention program to reduce test anxiety is 2~3 times a week for four weeks, with 20 min of any intensity aerobic exercise. In terms of intervention methods, the effect of exercise combined with psychotherapy is better than that of simple exercise intervention. In terms of intervention intensity, low-intensity intervention with an intervention frequency of >3 times a week and intervention period for >8 weeks provide the best effects. The effect of 31~60 min is the best in terms of the duration of a single intervention. The optimal exercise intervention program to reduce students’ test anxiety is to perform low-intensity aerobic exercise for more than eight weeks, more than three times a week, and 31~60 min each time. Only one study [19] reported the follow-up results after three months of intervention and showed that the effect remained positive. The rest of the studies did not report the follow-up results. More high-quality studies, especially high-quality interventional studies, need to be included in the follow-up to verify the reliability of the results.

## 5. Strengths and Limitations

Strengths of this meta-analysis: To our knowledge, this is the first meta-analysis of exercise intervention on test anxiety of students, including the impact of exercise intervention on test anxiety of primary, secondary, and college students. The evidence presented by the authors may be useful to school workers, policymakers, student mental health workers, students, and their parents and guardians by helping students overcome test anxiety.

This meta-analysis has certain limitations: There are relatively few randomized and non-randomized controlled trials of exercise intervention on students’ test anxiety, so the number of included studies is small. The quality of the studies was moderate, most of the randomization methods in the studies were unclear, and none of the articles mentioned allocation concealment. Only one article mentioned participant and performer blinding [25]. Due to the limitations of study search, the studies included in this meta-analysis are all published and available studies, and unpublished studies, including “grey studies”, were not included. These factors may have a certain negative impact on the results of the meta-analysis.

Second, the included studies used different outcome measure scales and the type, duration, and intensity of exercise interventions were not consistent across experiments. At times, the modality of exercise intervention was not uniform across the experiments (e.g., running, ball games, or cycling). All of these differences in interventions and the use of different scales in the studies will lead to heterogeneity between studies. Limited by the quantity and quality of included studies, the aforementioned conclusions need to be verified by more high-quality studies. Therefore, larger-sample, higher-quality RCTs need to be carried out in the future to increase the sample size to make the meta-analysis results more reliable.

## 6. Conclusions

The results of this meta-analysis show that exercise intervention can effectively alleviate students’ test anxiety. Performing 20 min of any aerobic exercise 2~3 times a week at any intensity for at least four weeks can reduce test anxiety levels. However, this is the minimum standard and the amount of exercise should not be limited to the minimum amount of exercise. Shorter, 10~15 min exercise interventions did not relieve students’ test anxiety. The optimal exercise intervention for relieving students’ test anxiety is to perform low-intensity aerobic exercise for 31~60 min more than three times a week for more than eight continuous weeks. Exercise combined with psychotherapy provides a better effect on reducing students’ test anxiety than simple exercise intervention. The main forms of aerobic exercise to relieve test anxiety are running, sports games, tai Chi and yoga, etc. The practical implications are that it will be useful to school workers, policymakers, student mental health workers, students, and their parents and guardians and help students overcome test anxiety. It is suggested that future studies should explore whether shorter intervention periods can effectively reduce students’ test anxiety, which can quickly and effectively reduce students’ test anxiety. Since the quality of the included study is generally low, the number of studies is small, and the demonstration strength is limited, the above conclusions still need to be verified by more high-quality RCTs.

## Figures and Tables

**Figure 1 ijerph-19-06709-f001:**
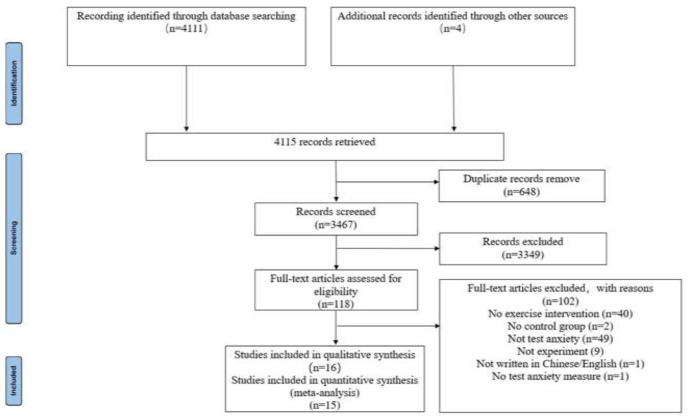
Study selection process.

**Figure 2 ijerph-19-06709-f002:**
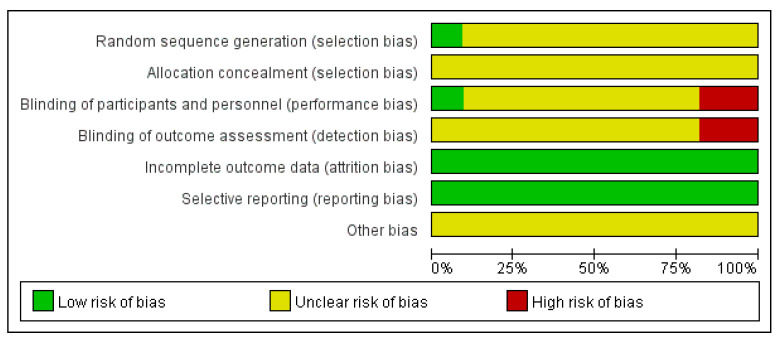
Review authors’ judgements about each risk of bias item presented as percentages across all RCT studies.

**Figure 3 ijerph-19-06709-f003:**
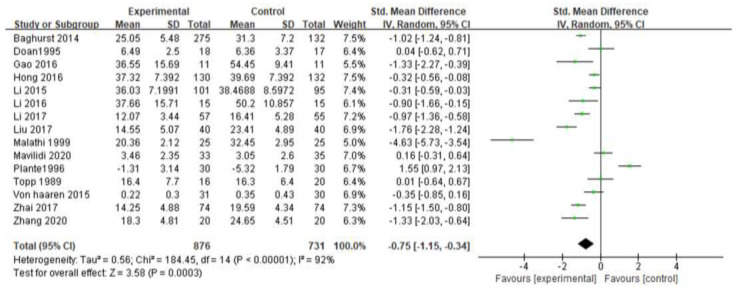
Forest plot of the effect of exercise intervention on test anxiety [17,18,19,20,22,23,24,25,26,27,28,29,30,31,32].

**Figure 4 ijerph-19-06709-f004:**
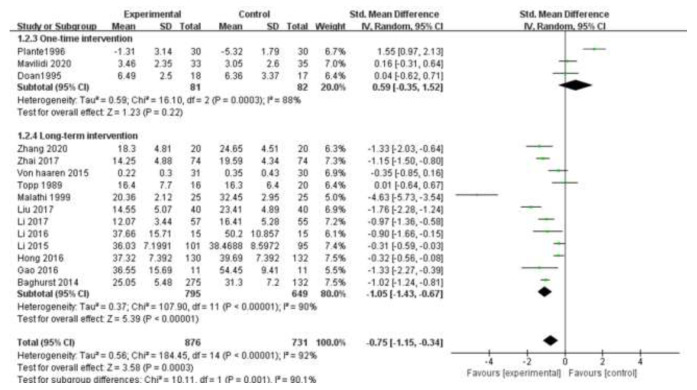
Subgroup analysis of intervention time [17,18,19,20,22,23,24,25,26,27,28,29,30,31,32].

**Figure 5 ijerph-19-06709-f005:**
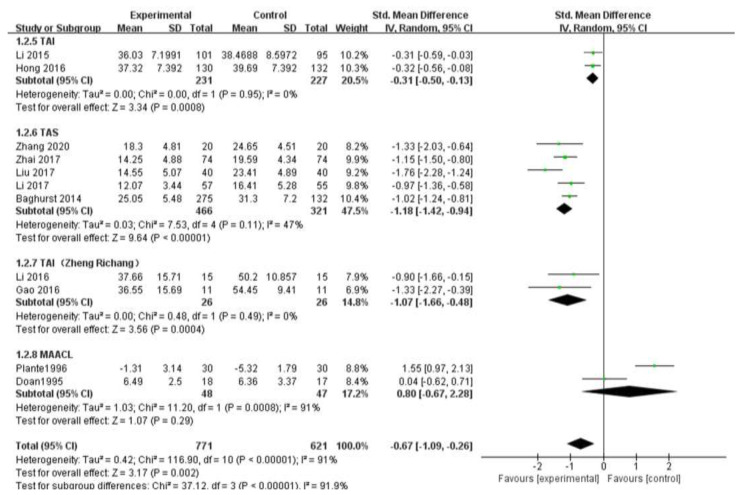
Subgroup analysis of outcome measurement scale [18,19,20,22,23,24,27,28,29,30,31].

**Figure 6 ijerph-19-06709-f006:**
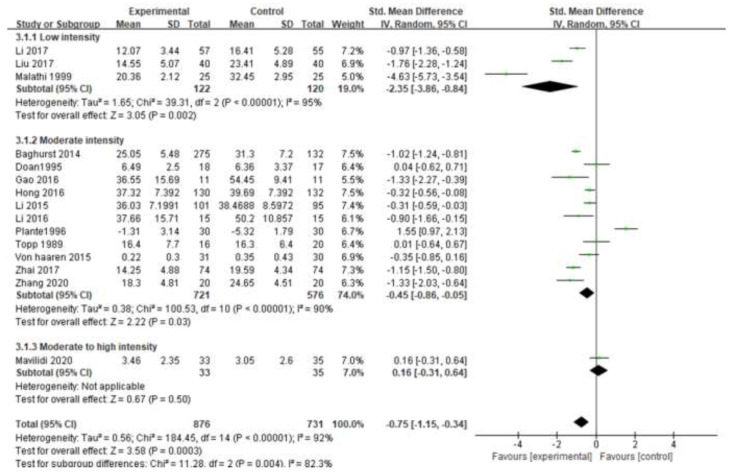
Subgroup analysis of exercise intensity [17,18,19,20,22,23,24,25,26,27,28,29,30,31,32].

**Figure 7 ijerph-19-06709-f007:**
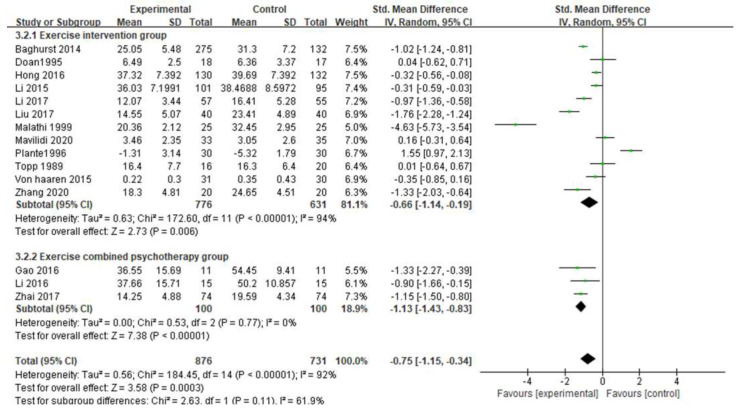
Subgroup analysis of intervention method [17,18,19,20,22,23,24,25,26,27,28,29,30,31,32].

**Figure 8 ijerph-19-06709-f008:**
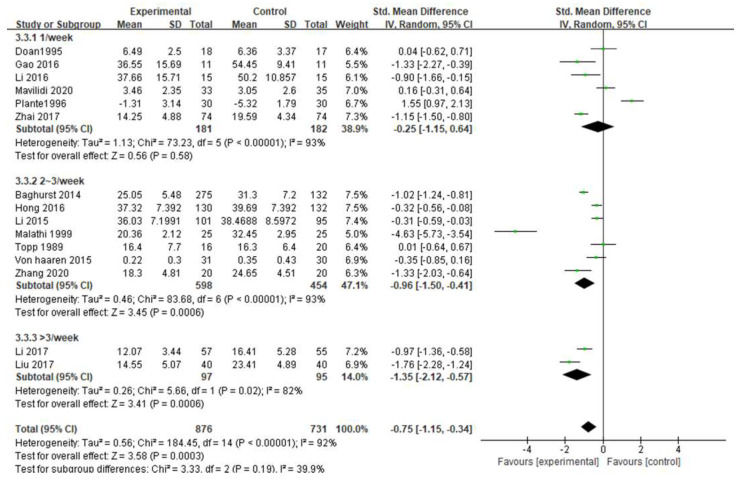
Subgroup analysis of exercise frequency [17,18,19,20,22,23,24,25,26,27,28,29,30,31,32].

**Figure 9 ijerph-19-06709-f009:**
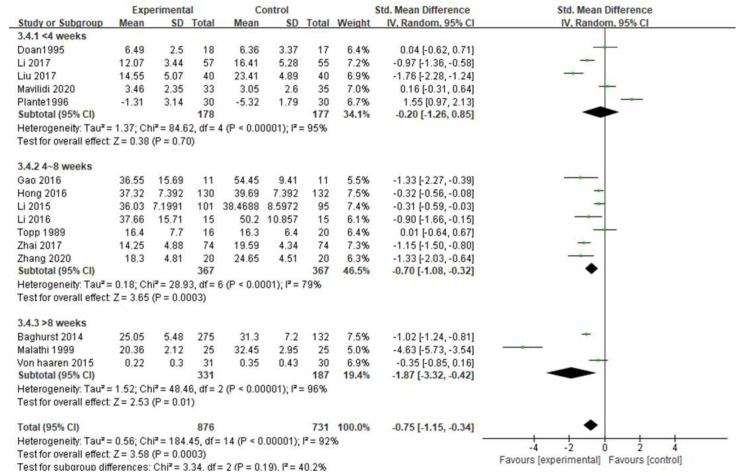
Subgroup analysis of intervention period [17,18,19,20,22,23,24,25,26,27,28,29,30,31,32].

**Figure 10 ijerph-19-06709-f010:**
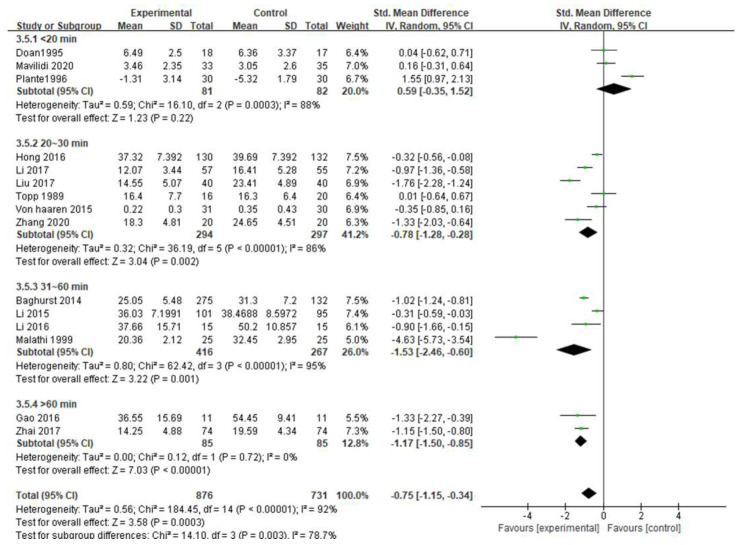
Subgroup analysis of exercise duration [17,18,19,20,22,23,24,25,26,27,28,29,30,31,32].

**Figure 11 ijerph-19-06709-f011:**
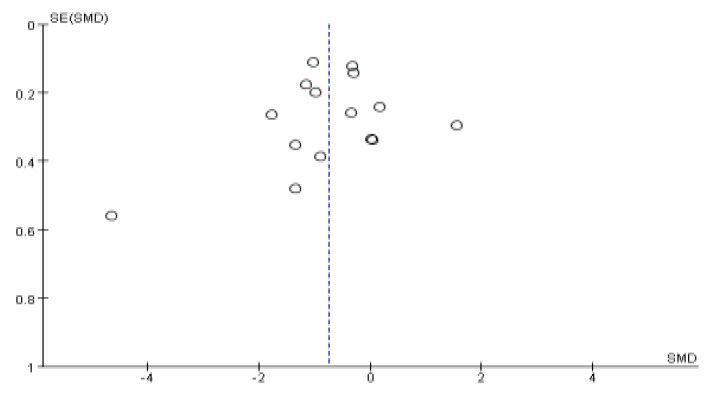
Funnel plot of publication bias.

**Table 1 ijerph-19-06709-t001:** Syntax sets used for searching PubMed.

#1	Population	Students OR Primary school students OR middle school students OR high school students OR college students OR adolescent OR Student OR School Enrollment OR Enrollment, School OR Enrollments, School OR School Enrollments
#2	Intervention	Exercise OR Sports OR Exercises OR Physical Activity OR Activities, Physical OR Activity, Physical OR Physical Activities OR Exercise, Physical OR Exercises, Physical OR Physical Exercise OR Physical Exercises OR Acute Exercise OR Acute Exercises OR Exercise, Acute OR Exercises, Acute OR Exercise, Isometric OR Exercises, Isometric OR Isometric Exercises OR Isometric Exercise OR Exercise, Aerobic OR Aerobic Exercise OR Aerobic Exercises OR Exercises, Aerobic OR Exercise Training OR Exercise Trainings OR Training, Exercise OR Trainings, Exercise OR Physical Education and Training OR Physical Education, Training OR Physical Education OR Education, Physical
#3	Outcomes	Test Anxiety OR Anxiety, Test OR Exam Anxiety OR Anxiety, Exam OR Exam Stress OR Stress, Exam OR Pre-Exam Anxiety OR Anxiety, Pre-Exam OR Pre Exam Anxiety OR Test fear
#4	#1 AND #2 AND #3	

**Table 2 ijerph-19-06709-t002:** The basic characteristics of included studies.

Authors (Year)/Country	Study Type	Sample Size		Intervention Program		Exercise Intensity	Dosage of Intervention (Time; Frequency; Duration)	Measuring Tools	Follow-Up
		Intervention Group	Control Group	Intervention Group	Control Group				
Mavilidi et al. (2020) /Australia [17]	RCT	33	35	Physical fitness	Word game	Medium and high intensity	10 min	Likert Scale (1–9)	No follow-up
Zhang and Zhang (2020)/China [29]	Non-RCT	20	20	Sports games	Blank	Predicted moderate intensity	20 min; 2/week; 7 weeks	TAS	No follow-up
Li and Li (2017)/China [18]	RCT	57	55	Tai chi	Blank	Predicted low intensity	30 min; 2/week; 2 weeks	TAS	No follow-up
Zhai et al. (2017)/China [19]	RCT	74	74	Sand tray combined with physical exercise	Blank	Moderate intensity	Sand: 120 min; 1/week PE: 70 min; 1/week; 8 weeks	TAS	3 months
Liu and Wang (2017)/China [20]	RCT	40	40	Tai chi	Blank	Predicted low intensity	30 min/day; 5 days	TAS	No follow-up
Thompson et al. (2016)/USA [21]	Cluster-RCT	359	432	PE lesson	Blank	Medium and high intensity	40 min	CTAS	No follow-up
Gao (2016)/ China [22]	RCT	11	11	Psychotherapy combined with physical exercise	Blank	Predicted moderate intensity	Psychotherapy: 80 min; 1/week PE: 20 min; 1/week 8 weeks	TAI	No follow-up
Li (2016)/ China [23]	RCT	15	15	Emotional therapy combined with sports games	Blank	Predicted medium and low intensity	Emotional therapy: 60 min; 1/week Sports games: 15–20 min; 1/week 8 weeks	TAI	No follow-up
Hong (2016)/China [30]	Non-RCT	130	132	Aerobic exercise	Blank	Predicted moderate intensity	30 min; 2/week; 2 months	TAI	No follow-up
Li et al. (2015)/China [24]	RCT	101	95	Jogging, sports games	Blank	Moderate intensity	30~40 min; 2~3/week; 1 month	TAI	No follow-up
von Haaren et al. (2015)/Germany [25]	RCT	31	30	Aerobic running	Blank	Predicted moderate intensity	30 min; 2/week; 20 weeks	NA	No follow-up
Baghurst and Kelley (2014)/USA [31]	Non-RCT	CV: 131 PA: 144	C: 132 SM:124	CV: Aerobic or anaerobic fitness PA: Ball game	C: Blank SM: Stress management	Predicted medium and high intensity	50 min; 3/week; 16 weeks	TAS	No follow-up
Malathi and Damodaran (1999)/India [26]	RCT	25	25	Yoga	Reading writing	Predicted low intensity	60 min; 3/week; 3 months	STAI	No follow-up
Plante et al. (1996)/USA [27]	RCT	30	30	Aerobic exercise	Watch videos	Predicted moderate intensity	15 min	TASC MAACL	No follow-up
Doan et al. (1995) /USA [28]	RCT	1.18 2.17	17	1. Cycling 2. Relaxation exercise	Reading magazines	Moderate intensity	15 min	TASC MAACL	No follow-up
Topp (1989) /USA [32]	Non-RCT	1.16 2.9	20	1. Aerobic dance 2. Relax group	Blank	Medium and high intensity	30 min; 3/week; 7 weeks	TAQ	No follow-up

Note: PA = physical activity; CV = cardiovascular fitness; SM = stress management; C = control group; TAI = Test Anxiety Inventory; TAS = Test Anxiety Scale; TAQ = Test Anxiety Questionnaire; MAACL = Multiple Affect Adjective Checklist; NA = Negative Affect; CTAS = Children’s Test Anxiety Scale; STAI = State Trait Anxiety Inventory.

**Table 3 ijerph-19-06709-t003:** Methodological Index for Non-randomized Studies (MINORS) scores for the included studies [29,30,31,32].

Item	Hong, 2016	Zhang and Zhang, 2020	Topp, 1989	Baghurst and Kelley, 2014
A clearly stated aim	2	2	2	2
Inclusion of consecutive patients	2	2	2	2
Prospective collection of data	2	2	2	2
Endpoints appropriate to the aim of the study	2	2	2	2
Unbiased assessment of the study endpoint	0	1	1	1
Follow-up period appropriate to the aim of the study	0	0	0	0
Loss to follow up less than 5%	0	0	0	0
Prospective calculation of the study size	0	0	0	0
*Additional criteria in the case of comparative studies*				
An adequate control group	2	2	2	2
Contemporary groups	2	2	2	2
Baseline equivalence of groups	2	2	2	2
Adequate statistical analyses	2	2	2	2
Final score	16	17	17	17
Maximum score	24	24	24	24

Note: Scores = 0 (not reported); 1 (reported but inadequate); 2 (reported and adequate). The total score is 24.

**Table 4 ijerph-19-06709-t004:** Results of subgroup analysis on the effect of exercise intervention on students’ test anxiety.

Moderator	Category	Number of Studies	SMD	CI	I^2^	Heterogeneity *p*
Intervention time	One-time	3	0.59	−0.35, 1.52	88%	0.0003
	Long-time	12	−1.05 **	−1.43, −0.67	90%	<0.00001
Measurement scale	TAQ	1	0.01	−0.64, 0.67	/	/
	NA	1	−0.35	−0.85, 0.16	/	/
	Likert Scale	1	0.16	−0.31, 0.64	/	/
	STAI	1	−4.63	−5.73, −3.54	/	/
	TAI	2	−0.31 **	−0.50, −0.13	0%	0.95
	TAS	5	−1.18 **	−1.42, −0.94	47%	0.11
	TAS(Zheng)	2	−1.07 **	−1.66, −0.48	0%	0.49
	MAACL	2	0.80	−0.67, 2.28	91%	0.0008
Exercise intensity	Low intensity	3	−2.35 **	−3.86, −0.84	95%	<0.00001
	Medium intensity	11	−0.45 *	−0.86, −0.05	90%	<0.00001
	Medium to high intensity	1	0.16	−0.31, 0.64	/	/
Intervention method	Exercise	12	−0.66 **	−1.14, −0.19	94%	<0.00001
	Exercise combined with psychology	3	−1.13 **	−1.43, −0.83	0%	0.77
Exercise frequency	1/week	6	−0.25	−1.15, 0.64	93%	<0.00001
	2–3/week	7	−0.96 **	−1.50, −0.41	93%	<0.00001
	˃3/week	2	−1.35 **	−2.12, −0.57	82%	0.02
Intervention period	<4 weeks	5	−0.20	−1.26, 0.85	95%	<0.00001
	4–8 weeks	7	−0.70 **	−1.08, −0.32	79%	<0.0001
	˃8 weeks	3	−1.87 *	−3.32, −0.42	96%	<0.00001
Exercise duration	<20 min	3	0.59	−0.35, 1.52	88%	0.0003
	20–30 min	6	−0.78 **	−1.28, −0.28	86%	<0.00001
	31–60 min	4	−1.53 **	−2.46, −0.60	95%	<0.00001
	˃60 min	2	−1.17 **	−1.50, −0.85	0%	0.72

Note: TAQ = Test Anxiety Questionnaire; NA = Negative Affect; STAI = State Trait Anxiety Inventory; TAI = Test Anxiety Inventory; TAS = Test Anxiety Scale; MAACL = Multiple Affect Adjective Checklist. **p* < 0.05; ***p* < 0.01.

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
