# Peer review of "Effects of Exercise Intervention on Students’ Test Anxiety: A Systematic Review with a Meta-Analysis"

_ijerph, 2022, doi:10.3390/ijerph19116709_

Round 1

Reviewer 1 Report

The manuscript presented a meta-analysis of the effect of exercise intervention on students´ test anxiety. Several studies, including reviews, are published from the topic. The topic is briefly described in a short Introduction referring to Chinese and international literature. Materials and methods are, on the contrary, described wider. However, it would be beneficial to know more about the study procedure: A) Why were exactly those databases chosen? B) Why was Chinese language included among English for the review? C) Why were students with depression, anxiety and other mental diseases excluded from the study (even though psychotherapy and emotion center therapy are mentioned in the Results section)? and D) Were the chosen studies peer-reviewed? This may also affect the validity of the meta-analysis. In Figure 1, additional records identified through other sources were also mentioned but not explained. What are these four studies?

Table 1 shows the syntax sets used for searching PubMed. An exhausting long list of syntax sets makes me wonder why ´exercise´, ´sports´ or ´physical activity´ were not only used to refer to interventions (in their many forms of writing) since the following search terms include these words (e.g., acute exercise, isometric exercise, aerobic exercise).

The strength of the manuscript is in describing and comparing the reviewed studies in detail. In Discussion, the authors compare successfully results from the meta-analysis and previous studies. However, some references went back to 1970s-1990s providing an older viewpoint on the previous studies. Few references for further research are provided.

In total, the manuscript needs proof reading in terms of the following aspects:

  • using spaces (e.g., lines 205-211 and in a reference list), full stops (e.g., lines 33-36, line 157, 466 eg or e.g., line 227, 439), and parenthesis (lines 160-162).
  • Please refer coherently to quantitative analysis (compare line 14)
  • In study limitations, consider the wide age spectrum of the students and its impact on the results.
  • Many sentences were not finished but, instead, resembled the notes (e.g., lines 8-9: The purpose is to …; line 108)
  • Table 1: In terms of clarity, outcomes should be separated from intervention.
  • Chapter 2.3.1: Text is written that the study will be done in the future. Please correct the text to a past tense. Please check the past tense throughout the manuscript when necessary.
  • Line 178: What do you mean with ´all genders´?
  • Table 3: Level the text in the ´Moderator´ column to the top of the rows which will ease the interpretation of the Table. E.g., Intervention time and One-time in the same row.
  • Consider replacing ´literature´ with ´study´ throughout the text.
  • Discussion refers to a result of the meta-analysis which is not described in the Results section: ´3 of the 15 studies reported heart rate during exercise´. This result should be described in the Results section.
  • Discussion would be stronger when providing more references to earlier studies. For example, line 378: ´Some studies have also proved this´ (which studies?), lines 411-415 and 422-424.

Author Response

Thank you very much for your comments. It would be of great help to me. Please see the attachment.

Reviewer 2 Report

Title: Authors may consider if the word “test” is really necessary.

Abstract: In the first sentence, authors should review a possible missing like “The aim of this research was”. In lines 12 and 29, I suggest that authors do not use abbreviations in abstract but in the following sections.

Keywords: So as to foster the visibility of this paper in the different dtabases this journal is indexed in, I recommend that keywords are not repeated in title.

Introduction: Punctuation marks must be reviewed in the first paragraph. In line 45, authors affirm that “previous meta-analysis studies have (…) but did not involve exercise intervention [7]). The reference number 7 is from 2003, that is, very old. It is probable that later meta-analysis did include exercise interventions. I wonder why authors talk about Chinese student instead of students from any countries with international research as background.

Searh strategy: Why these databases were chosen? What criterias were followed with this regard? Authors may consider if information provided in lines 70-72 is appropriate to be included in this section or in introduction. I suggest that information from line 81 to 86 should be in a different paragraph.

In section 2.3.1 Data extraction and processing: Ensure that the use of verbal times is correct.

In 3.1. Section Outcomes: Numbers from lines 142 to 145 do not coincide with figure 1. Review redaction in line 156 and 159 and the use of spaces in lines 160 and 162.

In 3.2.3. Measurement of study results: Although this information could be deduced, in my view, the result that only two studies used more than one instrument to assess anxiety is also relevant and could be remarked.

In 3.3. Assessment of the risks of literatura bias, I would like to know how these risks were calculated. Why the information included in table 2 is not also provided for RCT studies? Could all names from figure 3 be written with non-chinese characters? This would help other country researchers. Authors must increase resolution of figures 4-10. Now, it is really hard to read their content.

In Discussion: What studies “suggest that 40-60 minutes work better” to produce psychological benefits? (line 351). More studies should be considered to discuss this research findings and, when possible, try to use more recent studies.

In Strengths and limitations: The different ways to consider that one exercise has low, moderate or high intensity among the studies include in this research make difficult to compare them.

In Conclusions: In my humble view, more future research lines (not only the one included in lines 481-483) would be interesting for readers.

Author Response

(The authors gave the same response as above.)

Reviewer 3 Report

The abstract is very extensive and lacks simplicity.Moreover, the mentioned section contains statements that are repeated twice. The results in the summary must be presented more clearly, so that readers can better understand the main findings of your study. On the other hand, introduction is really short. The authors did not define basic research concepts such as anxiety. The readership may have difficulty understanding the entire meta-analysis due to inadequate familiarity with the elementary terms of this study. Additionally, lines 42-45 – this statement should be supported by more references. The main criticism in the methods section relates to the criteria for including and excluding literature that are practically the same.

Can you explain the purpose of Caspersen' s definition of physical activity, please? The fourth criterion for including studies is also unreasonable.

Futhermore, in the context of the literature search strategy, the keywords are presented in an inappropriate way. The authors did not present search strategy with "AND". Generally, my recommendation is to submit Table 1 as ansupplementary file.

Lines 89-94 – mentioned text does not belong to the section for data extraction and is more related to the literature search strategy.

Significant corrections will also be required in the resultssection. In Figure 1, check again the number of studies you excluded with reasons, for each item, please. Since the title of you meta-analysis is the impact of exercise interventions on students' anxiety test, why did not you removed studies that combined exercise with psychotherapy? In terms of determining the intensity of exercise, how did you define the intensity of training in studies in which it was not reported, did you have any objective criteria or? In table 2, the research should be presented in accordance with the year of publications, starting with those recently published. It is necessary to check the whole table again!

Is the subtitle "Study design" consistent with the text below it?

Lines 192-193 - Please report the purpose of this statement!

In Table 3, the "Category" segment is currently confusing and will be unclear to readers.

Lines 268-271 – Please check the whole text once more.

Are you sure that the maximum values of the effect size from the figures are shown in the text in the sections exercise frequency, intervention period and exercise duration?

The discussion is very similar to the results. There is an obvious lack of comparisons with other studies, as well as explanations of the results obtained. In addition, about 10 statements were observed which must be supported by references. Moreover, this section contains a large number of sentences that are quite confusing and need to be presented more simply. Check thoroughly once again all the paragraphs in the discussion, please.

Please list the strengths of your research in the "Strengthsand limitations" section.

Lines 459-461 - Why do you think this claim is a limitation of the presented study?

Study lack practical implications. The presented findings unequivocally show that exercise interventions reduce the level of students' test anxiety. Does this mean that students will be more efficient in taking tests? The authors should emphasize the practical implications of the study in manuscript. It is important to add that all sections of this research (abstract, introduction, methods, results and discussion) contain unclear sentences in which a large number of grammatical mistakes are observed. Researchers must conslut an English language specialist before resubmitting the manuscript. Many non-scientific terms were also usedin the study. Authors must pay attention to the manner of expression and style of writing.

Author Response

(The authors gave the same response as above.)

Round 2

Reviewer 1 Report

Hi, thank you for modifying the manuscript. Now it is a better read. However, I thought that my comments will be answered in the manuscript (not only in a response letter) if the other readers are also interested in knowing more about the research. Explanations would also strengthen the scientific value of the manuscript. For finalizing the manuscript, please check the hyphens (e.g., line 36 = pro-duced) and layout of Table 2. Good luck for your research. 

Author Response

Thank you very much for your comments. We have included the explanations of your comments in the manuscript. This will make it easier for the reader to understand. We have modified the hyphens and layout of Table 2.

Thank you again for your good wishes.

Reviewer 2 Report

Dear authors,

I would like to congratulate you for your effort since this new version of the paper is, in my humble view, much better than the previous one.

However, there are some mistakes in references section. Review every reference but, for instance, the use of [] tends to be wrong.

Author Response

Thank you for your careful review. Your comments was of great help to me. We reviewed the references again and deleted "[]". We also found the ", "that we use when spacing two authors. Now we have changed to ";". Thank you again for your comments. Have a nice day.

Reviewer 3 Report

The authors have partially fulfilled the previous remarks and comments. Additionally, all changes in the text must be marked with a track change. The following observations need to be analyzed again:  

The abstract is clearer for readers, but the introduction is really short again.

The inclusion and exclusion criteria now look better, but the literature search strategy is still the same. Please, justify yourself!

Lines 91-96: Why did you ignore the previous comment regarding this text?!

What about the previous comment related to determining the intensity of the exercise?

The entire Table 2 should be shown on one page. It will be clearer for the readership.

Check again the following comments, please:

-          Is the subtitle "Study design" consistent with the text below it?

-          Lines 196-197: Please report the purpose of this statement

-          Lines 268-271 – What did the subgroup analysis show? Please, explain! 

Changes in the discussion section are also not visible. There are still a few statements that need to be supported by references!

Why did you mention the practical implications of the study in the introduction? They should be presented as part of the discussion, conclusion, or as a separate section.

Numerous grammatical mistakes, confusing sentences, and unscientific terms are still observed in the manuscript. Researchers must consult an English language specialist before resubmitting the manuscript.